# HetroTraffSim: A Macroscopic Heterogeneous Traffic Flow Simulator for Road Bottlenecks

Ali Zeb [1], Khurram S. Khattak [1,2,*], Muhammad Rehmat Ullah [1,2], Zawar H. Khan [1,3] and Thomas Aaron Gulliver [3]

1   National Center for Big Data and Cloud Computing (NCBC), UET, Peshawar 25000, Pakistan
2   Department of Computer Systems Engineering, UET, Peshawar 25000, Pakistan
3   Department of Electrical and Computer Engineering, University of Victoria, Victoria, BC V8W 2Y2, Canada
*   Correspondence: khurram.s.khattak@gmail.com; Tel.: +92-348-0006688

**Abstract:** Smart mobility is crucial for future smart cities. Traffic simulation software (TSS) is an important tool for efficient planning and management of road networks to achieve this goal. Many TSS tools have been developed for both microscopic and macroscopic homogenous traffic flow. However, only two (SUMO and HetroSim) are applicable to heterogeneous traffic. In this paper, HetroTraffSim is proposed to simulate macroscopic heterogeneous traffic flows at road bottlenecks. It is developed using the Unity3D engine and is based on a second-order traffic flow model. It is evaluated for a 360 m road segment on University Road, Peshawar, Pakistan. This segment contains a 78.5 m bottleneck which causes traffic congestion. This bottleneck is due to the construction of a pedestrian overhead bridge which reduces the three-lane road to two lanes. HetroTraffSim provides normalized traffic velocity, average traffic density, traffic flow, and time, as well as the temporal and spatial evolution of traffic. The results obtained show that a change in the distance headway affects the traffic flow, velocity, and density. Further, HetroTraffSim can be extended to automated traffic flows using raycasting. It can easily be used to create realistic traffic scenarios and the computational complexity is low due to the small-degree polynomials employed. HetroTraffSim can be used by traffic planners to improve traffic flow and public safety.

**Keywords:** traffic simulation; simulations software; heterogeneous traffic; road bottleneck; lane change; Unity3D; raycasting

## 1. Introduction

It is estimated that the world urban population will increase by up to 68% by 2050 [1,2]. Thus, urban mobility is emerging as a challenge for future smart cities. The associated problems include traffic congestion, increased pollution, increased accidents, reduced productivity, and degradation in quality of life. In large American cities, it is estimated that traffic congestion will cost USD 489 billion from 2017 to 2027. This cost is based on wait times, productivity losses, fuel consumption, and carbon emissions [3]. Furthermore, road injuries are predicted to cost USD 1.8 trillion between 2015 and 2030 [4]. Thus, effective design, planning, and management of urban road networks are crucial for the success of smart cities.

Traffic density is the number of vehicles per unit length, and velocity is the distance traveled in a unit of time. Traffic is at equilibrium when the velocity is maintained based on the density. Traffic flow is the product of density and velocity. Homogenous traffic is categorized by strict lane discipline and equilibrium conditions are followed. Heterogeneous traffic does not follow lane discipline. The latent distance headway is the distance between adjacent vehicles. During congestion, this headway is reduced. Lane change frequency increases as the latent distance headway decreases, resulting in larger variations in flow and often congestion. In this case, road infrastructure is not effectively utilized.

Microscopic traffic characterization considers individual vehicles and is based on probabilistic parameters. Conversely, macroscopic traffic characterization is based on

average traffic parameters which are deterministic. It is used to examine average traffic behavior, i.e., aggregate velocity, density, and flow. At critical density, the flow is maximum. Beyond this density, the flow decreases and congestion can occur. Congestion results in excessive acceleration and deceleration which increases travel time and pollution.

Traffic Simulation Software (TSS) is an important tool for efficient road network management. It is used to simulate vehicle interactions based on the distance between vehicles, the time required to align to forward conditions, driver response time, and the time required for a lane change. TSS such as Paramics, PTV VISSIM, AimSum, CORSIM, MITSIM consider homogeneous flow [5,6]. PTV VISSIM uses a psychophysical driver model which is based on an extensive examination of traffic parameters. AimSum employs Artificial Intelligence (AI) and characterizes traffic based on vehicle type and travel time. It has been used to reduce congestion considering the turning ratios at intersections. CORSIM combines NETwork SIMulation (NETSIM) for urban road simulation and FREeway SIMulation (FRESIM) for highway simulation. It provides microscopic traffic characterization and considers driver behavior. MITSIM also provides microscopic traffic characterization and is based on driver behavior. CORSIM was developed to examine traffic flow under breakdown and recovery conditions. It is not applicable to heterogeneous traffic and congestion.

Few TSS tools have been developed for macroscopic traffic characterization [5]. HetroSim considers small road segments and supports only a few vehicle types, sizes, and speeds. SUMO can estimate homogenous traffic on large road networks but ignores lanes. Traffic flow in most developing nations is heterogeneous and lane discipline is often ignored as governments lack the resources for enforcement. Therefore, a macroscopic heterogeneous traffic simulator is required to predict traffic flow for these conditions.

In this paper, HetroTraffSim for macroscopic heterogeneous traffic flow simulation is presented. This tool considers lane changes. It has been developed using the Unity3D engine and is based on a second-order partial differential system [7]. To the best of our knowledge, this is the first tool that can simulate macroscopic heterogeneous traffic at road bottlenecks. HetroTraffSim has the following advantages over existing tools.

- Macroscopic heterogeneous traffic flow can be simulated at road bottlenecks.
- The user-friendly Graphical User Interface (GUI) can be used to create realistic traffic scenarios using parameters such as road width and length, number and types of vehicles, time and distance headways, and minimum and maximum velocities.
- It can be used to predict traffic density, average velocity, and average flow temporally and spatially.
- The critical density of a road can be determined.

The remainder of this paper is organized as follows. The related work is presented in Section 2. Section 3 introduces the proposed simulation software framework. HetroTraffSim is evaluated in Section 4 and a discussion of the results is provided in Section 5. Finally, Section 6 provides some conclusions and suggestions for future research.

## 2. Related Work

Ye Li et al. [8] developed Cooperative Adaptive Cruise Control (CACC) and Variable Speed Limit (VSL) techniques to decrease bottlenecks and rear-end collisions on freeways. They considered microscopic traffic characterization using Time Exposed Time-to-collision (TET) and Time Integrated Time-to-collision (TIT). Three vehicle models were employed, namely, CACC path, Adaptive Cruise Control (ACC), and the Intelligent Driver Model (IDM). It was shown that combining CACC and VSL improves efficiency by 33% and reduces rear-end collisions by up to 98%. Further, both the efficiency and safety are better than with just VSL control. They also showed that integrating CACC and VSL control with a Vehicle to Infrastructure (V2I) system can improve efficiency and reduce rear-end collisions at bottlenecks.

Song and Sun [9] investigated a congested freeway on-ramp bottleneck in Shanghai, China. They considered microscopic traffic characterization using VISSIM for traffic prediction and congestion mitigation at a bottleneck. Nine vehicle parameters were employed

and the results were validated using a real dataset. However, the implementation was complex and it was difficult and costly to obtain the data.

Raju et al. [10] investigated a bottleneck on a multi-lane road using VISSIM. The upstream road was divided into seven segments to analyze the trajectories of six types of vehicles. It was shown that upstream traffic has a significant effect on the bottleneck.

Qu et al. [11] explored traffic congestion on freeways using VISSUM. Spectral clustering was used to determine the transition boundary between free flow traffic and congested traffic. The results obtained can be employed to control traffic for efficient flow. Rahimi et al. [12] evaluated AIMSUN and SimTraffic on four different roads in northern Iran. Although both tools could identify bottlenecks, AIMSUN provided more accurate speeds, flows, and distances.

Alghamdi et al. [13] presented a comprehensive review of traffic prediction models and simulation tools. The most commonly employed non-parametric prediction models were examined. Kim et al. [14] developed Corridor Simulator (CORSIM) for microscopic traffic characterization which is based on FRESIM and NETSIM.

In [15,16], SUMO was used to examine congestion in heterogeneous traffic. Malik et al. [17] proposed a smart traffic system for routing emergency vehicles to avoid congested road segments in Pakistan. Jayasinghe et al. [15] presented a framework for the automated calibration of lane change and car-following model parameters for heterogeneous traffic. The calibration was performed using the Simultaneous Perturbation Stochastic Approximation (SPSA) with SUMO. Their framework was applied to large road networks in Sri Lanka. Can et al. [16] used SUMO to study the impact of congestion arising from random incidents in Hanoi, Vietnam, on travel time. It was shown that travel time and congestion intensity are proportional to incident duration.

## 3. HetroTraffSim Software Framework

This section presents HetroTraffSim, a simulator for heterogeneous traffic flow at road bottlenecks. It was developed using the Unity3D engine and the C# language. This engine was chosen because it offers a fast, efficient, and realistic simulation development environment. It provides cross-platform functionality enabling developers to build software for different platforms such as Android, iPhone, and Windows. The HetroTraffSim framework flow diagram is shown in Figure 1. The GUI allows users to input traffic parameters such as road length and width, bottleneck length, density, number and types of vehicles, minimum and maximum velocity, and simulation time. HetroTraffSim simulates traffic flow using a second-order macroscopic traffic flow model and real roadside traffic data. Traffic parameters such as flow, density, and velocity are provided for analysis and stored in the Google cloud. The components of the HetroTraffSim framework are described below.

### 3.1. Road Infrastructure

HetroTraffSim allows users to specify road infrastructure parameters such as road length and width, as well as bottleneck length and width. Further, several road topologies are available such as straight, bottleneck, intersection, roundabout, U-turn, circular, and T-section, as shown in Figure 2. The 3D road layout is created using the Easyroad3D package in Unity3D. Several environments are available such as country, urban, and mountain. Moreover, Blender software is employed to create 3D objects such as buildings, road signs, roadside fences, bus stations, petrol pumps, and flyovers, to obtain a realistic simulation environment as explained in Section 3.2. A bottleneck of length 78.5 m is considered in Section 4 to evaluate HetroTraffSim.

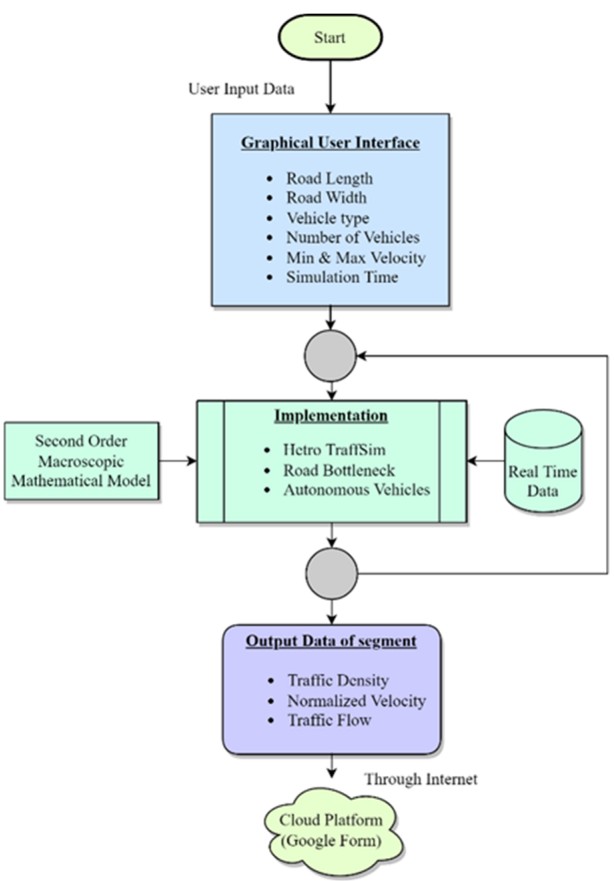

**Figure 1.** The HetroTraffSim framework flow diagram.

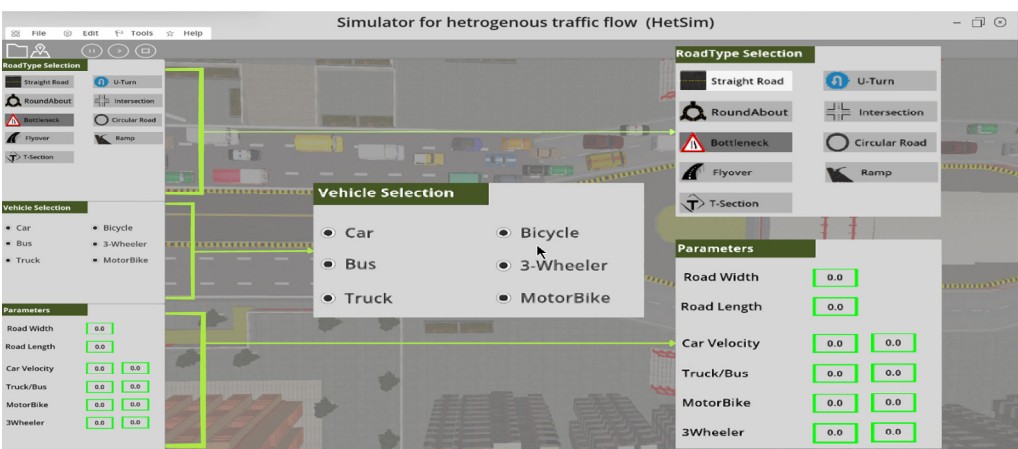

**Figure 2.** The HetroTraffSim GUI for road infrastructure showing local conditions in Peshawar, Pakistan.

### 3.2. Vehicular Flow

For realistic traffic simulation according to local traffic conditions, HetroTraffSim allows users to select vehicle and road types as shown in Figure 2. The dimensions of vehicles on Pakistan roads available in HetroTraffSim are provided in Table 1 [18]. The maximum velocity for each vehicle type can be set according to real traffic conditions.

**Table 1.** Vehicle types with dimensions and maximum velocities.

| Number | Type | Dimensions (Length, Width, Height) | Maximum Velocity |
|--------|------|-----------------------------------|------------------|
| 1 | Taxi | 4.5 m, 2.0 m, 1.8 m | 10 m/s |
| 2 | Sports car | 4.7 m, 1.9 m, 1.4 m | 10 m/s |
| 3 | Bus | 7.6 m, 2.3 m, 3.1 m | 9 m/s |
| 4 | Van | 4.9 m, 1.95 m, 1.8 m | 10 m/s |
| 6 | Ambulance | 5.3 m, 2.2 m, 2.4 m | 9 m/s |
| 7 | Police car | 4.6 m, 1.85 m, 1.7 m | 10 m/s |
| 8 | Small truck | 6.2 m, 2.46 m, 2.6 m | 8 m/s |

For traffic generation, an instantiation object is placed at the start of each lane, as explained in Section 4.1. These objects are used to initialize the following vehicle parameters.

- Maximum and minimum delay (the minimum and maximum arrival rate of vehicles to a road section).
- Spawning time (between minimum and maximum delay).
- Spawning vehicle type (as in Table 1).
- Instantiation number (to differentiate between agents).

Each vehicle is assigned a random path during instantiation to follow from the initial location to the destination. The vehicles (agents) in the simulation area are displayed on the main canvas with their speeds as shown in Figure 3.

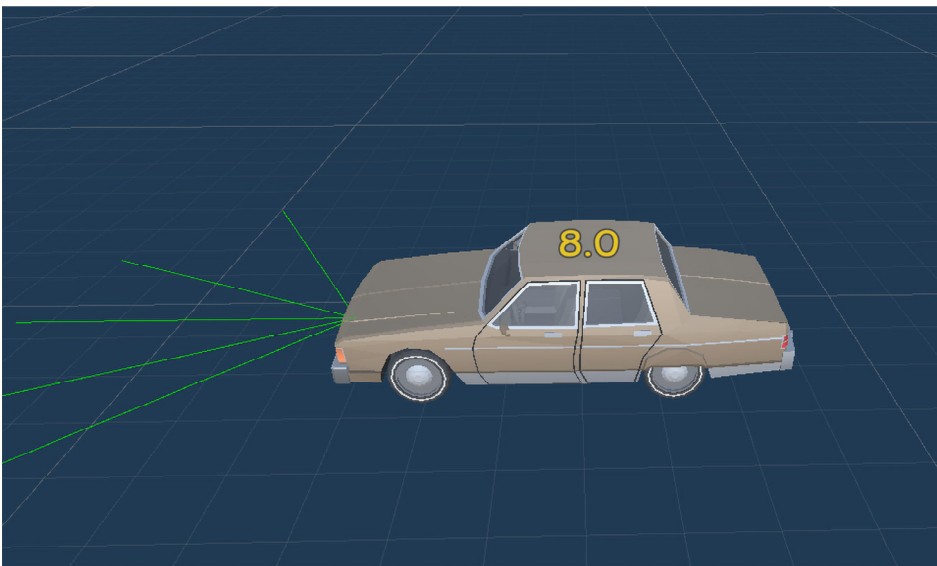

**Figure 3.** A vehicle (agent) with raycast lines and speed.

An AI script for raycast lines is used for vehicle maneuvering and collision avoidance. This is enhanced using three components of Unity3D with in-vehicle sensors, as described below.

- NavMesh Surface is used by the vehicle to distinguish between drivable and nondrivable areas [19]. The surface properties are editable to guide the AI to follow user-generated paths. For traffic flow simulation in this paper, only one road direction is considered (the right side in Figure 4a).

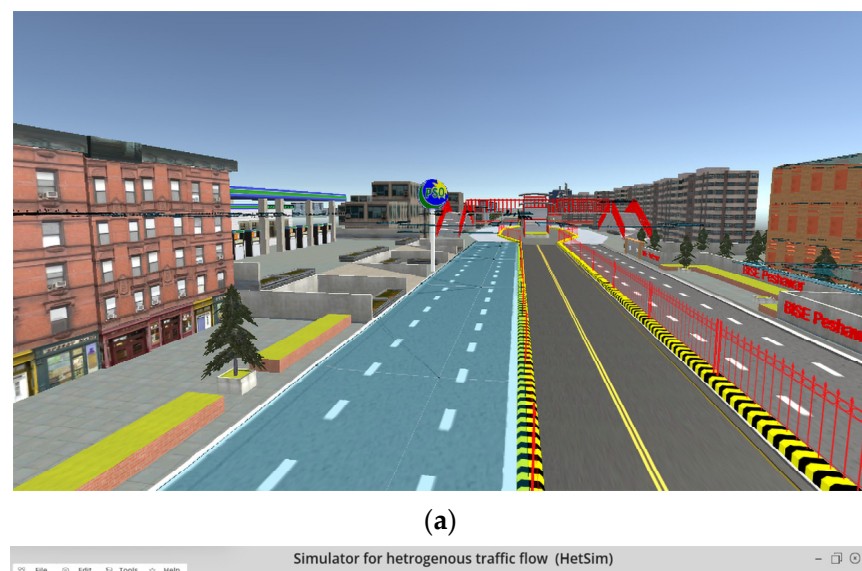

(**a**)

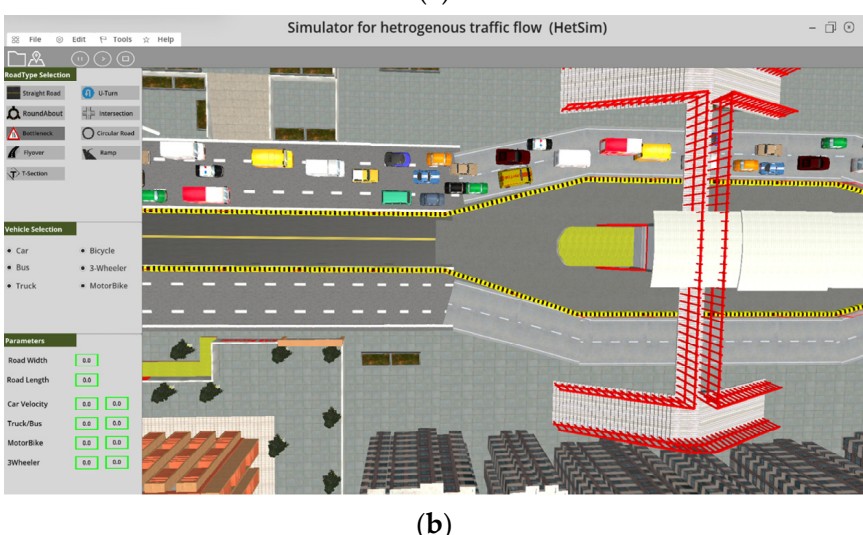

(**b**)

**Figure 4.** (**a**) NavMesh Surface for the simulated road and (**b**) simulated traffic on this surface.

- A NavMesh agent provides agent (vehicle) road maneuvering and obstacle avoidance capabilities. The agent maneuvers on the NavMesh surface and the NavMesh agent allows vehicles to find appropriate paths.
- A NavMesh obstacle is attached to each obstacle for avoidance purposes. If it is a stationary obstacle then the NavMesh agent will avoid it, and if it is a moving obstacle a hole will be created in the NavMesh surface which can be updated for the agent to avoid.

### 3.3. Macroscopic Traffic Flow Model

HetroTraffSim uses the Khan–Gulliver (KG) model [7] to simulate macroscopic heterogeneous traffic flow. This model characterizes vehicle alignment according to the forward conditions. The parameters used for traffic alignment are safe time ($t_s$), safe distance ($d_s$), maximum normalized density ($\rho_m$), and maximum normalized velocity ($v_m$), as given in Table 2. Safe time is the sum of reaction time (the time to react to forward traffic) and transition time (the time to align to forward traffic).

**Table 2.** Khan–Gulliver (KG) model parameters.

| Name | Parameter | Value |
|---|---|---|
| Safe time | $t_s$ | 4 s |
| Safe distance | $d_s$ | 1.2 m |
| Maximum normalized density | $\rho_m$ | 1 |
| Maximum normalized velocity | $v_m$ | 10 m/s |
| Traffic density | $\rho$ | Input |
| Traffic flow | $v\rho$ | Runtime |
| Equilibrium velocity distribution | $v(\rho)$ | Runtime |
| Transition distance | $d_{tr} = d_s + v_m t_s$ | 41.2 m |
| Relaxation time | $T$ | 5 s |
| Road section length | | 360 m |
| Number of road segments | | 12 |
| Segment length | | 30 m |

The KG model is more realistic than other models because it is based on traffic physics. The distance headway is covered by vehicles to align with forward traffic. Parameters such as the difference between forward and rearward traffic conditions and distance headway to align are considered in the KG model. With the Payne–Whitham model, traffic behavior is characterized using a constant for driver presumption [20], which is unrealistic. The Zhang model is based on the equilibrium velocity distribution [21] and so cannot characterize traffic at large changes. As a result, the velocity can go beyond the upper and lower bounds. Aw and Rascle characterized traffic based on density [22]. According to this model, traffic changes are based on a constant exponent considering fluid behavior, so traffic physics is ignored.

There are five forward raycast sensors as shown in Figure 3. These are employed for forward vehicle detection. These sensor values and the KG model parameters in Table 2 are used for vehicle alignment with forward traffic.

## 4. Simulation and Results

### 4.1. Simulation Setup

As a metropolitan center and provincial capital, urban congestion is a major challenge for Peshawar, Pakistan. To reduce this congestion, a Bus Rapid Transit (BRT) project was initiated. This project consists primarily of laying a two-lane road in the middle of major arterial roads exclusively for buses as shown in Figure 5. Thirty-one BRT stations serve as bus stops and provide overhead pedestrian bridges for crossing the road. The piers for these bridges reduce the three-lane road to a two-lane road. To evaluate the impact on traffic flow, a 360 m section of University Road, Peshawar, Pakistan, is considered. It spans from Islamia College (33.99819° N, 71.4754° E) to Board Bazar (33.99782° N, 71.46972° E), as shown in Figure 5. The BRT station (Board Bazar) is at (33.99808° N, 71.47231° E). There are three lanes, 13 m wide, before and after the pedestrian bridge. This is reduced to two lanes, 8 m wide, between the piers. This 5 m reduction in width results in a bottleneck.

The initial velocity, traffic flow, and density were recorded at the BRT station for use in HetroTraffSim. Then the evolution of heterogeneous traffic at the bottleneck is simulated. For traffic flow analysis, 12 road segments, 30 m in length, on the road section are considered as shown in Figure 6. Note that driving in Pakistan is on the left side of the road. Each of these segments is further divided into three subsegments ($i$, $i + 1$, $i + 2$) 10 m in length as indicated in Figure 6. Actual roadside traffic data are essential for realistic traffic flow simulation. Thus, traffic flow parameters were recorded for each subsegment and averaged in each segment for better results. The road section was observed from Monday, December 23, 2019, to Friday, December 27, 2019, and from 08:40 a.m. to 5:20 p.m. each day [23]. During these times, speeds in the bottleneck varied from 4 km/h to 26 km/h.

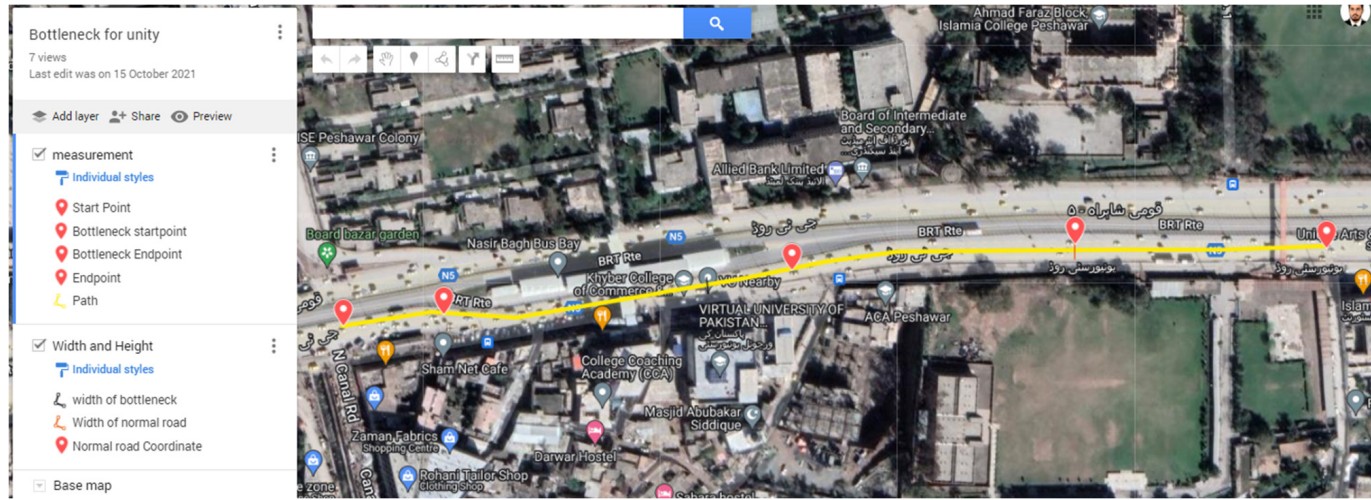

**Figure 5.** The road section with the BRT station selected for simulation.

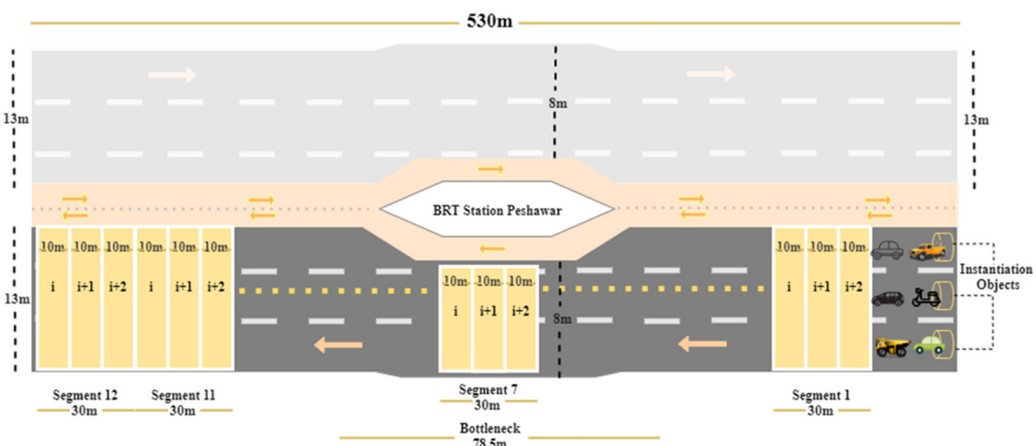

**Figure 6.** The 360 m road section with a bottleneck divided into segments in HetroTraffSim.

For HetroTraffSim evaluation, traffic flow on the road was simulated for 5 s. A total of 180 vehicles were randomly instantiated between 0.5 s and 1.3 s using three instantiation objects as explained in Section 3.2 and shown in Figure 6. Each object generated 60 to 75 vehicles in each lane according to the spawning time determined by local traffic conditions. The vehicle types generated are detailed in Table 1. The KG model was implemented using the Roe decomposition scheme [7].

### 4.2. Spatial and Temporal Traffic Normalized Density

The normalized traffic density on the road segment obtained using HetroTraffSim is shown in Figure 7. At 1 s, the density is 0.20 in segments 3 to 6. In segments 1, 2, and 9, the density is 0.10, but in segment 7 it is 0.15. The density is 0.20 in segments 10–12. At 2 s, the density in segments 1 and 2 is 0.20, and increases to 0.29 in segments 3, 4, 5, and 7. In segment 6, the density is 0.25. The density in segment 8 is 0.21 and increases to 0.23 in segment 9. The density decreases from 0.26 in segment 10 to 0.21 in segment 12.

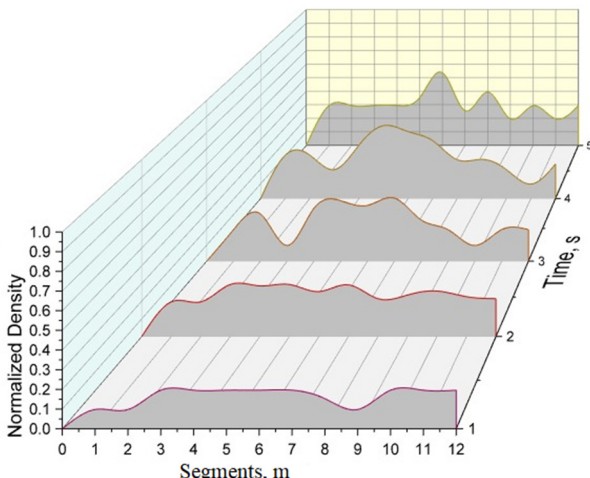

**Figure 7.** Average normalized density over time and space on the road section.

At 3 s, the lowest density is 0.10 in segments 3 and 10 while the highest density is 0.39 in segments 5 and 7. In segments 1, 9, 11, and 12, the density is 0.20. In segment 2, it is 0.29 while in segment 6 it is 0.35. In segment 8, it is 0.26. At 4 s, the density is approximately 0.29 in the first two segments and decreases to 0.20 in segment 3. It increases in segments 4 and 5 to 0.49 and then decreases in segments 6, 7, and 8 to 0.26. It is 0.10 in segment 11 and 0.23 in segment 12. At 5 s, the density is approximately 0.29 in the first four segments. It then increases in segments 5 and 6 to a maximum of 0.54. The density then decreases and is a minimum of 0.2 in segments 9 and 11.

### 4.3. Spatial and Temporal Traffic Flow

Figure 8 presents the spatial and temporal traffic flow in the road section. At 1 s, the flow is approximately 0.98 veh/s in segments 1 to 4. It rapidly increases to 2.89 veh/s in segment 5, decreases in segments 6–8, and increases again to approximately 2.90 veh/s in segments 9–12. At 2 s, the flow is approximately 1.94 veh/s in segments 1–4 and 9–11. In segment 6, the flow is lowest at 0.99 veh/s, while in segment 8 it is highest at 3.65 veh/s. At 3 s, the flow is approximately 1.95 veh/s in segments 1–5 and 1.90 veh/s in segments 9–12. The largest flow is 2.92 veh/s in segment 7, while the smallest flow is 0.97 veh/s in segment 8. At 4 s, the flow varies in segments 1–5. In segment 5, it is 2.89 veh/s and in segment 3 it is 1.95 veh/s. In segment 7, the flow is lowest at 0.99 veh/s and in segment 11, it is highest at 3.85 veh/s. At 5 s, the flow is approximately 2.90 veh/s in segments 1–4 and 5–7. In segment 8, it is 2.91 veh/s, in segment 11 it is 2.60 veh/s, and it is lowest at 0.99 veh/s in segment 10.

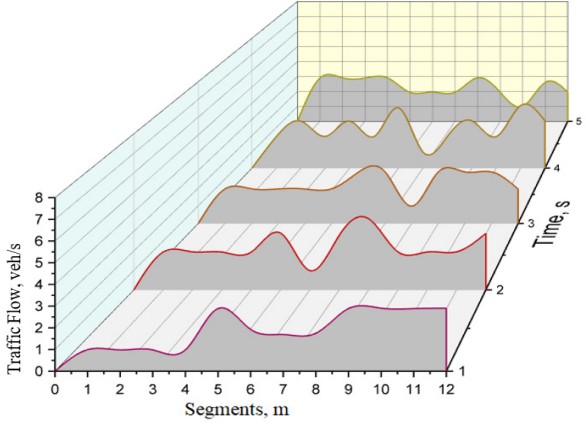

**Figure 8.** Average traffic flow over distance and time.

### 4.4. Spatial and Temporal Normalized Traffic Velocity

The spatial and temporal normalized traffic velocity on the road section ranges from 0 m/s to 10 m/s as shown in Figure 9, and is smallest before and within the bottleneck. At 1 s, the velocity is 9.2 m/s in segment 1. The velocity decreases to 7.7 m/s in segment 9 and then increases to 9.1 m/s in segment 11. At 2 s, the velocity in segment 2 is 9.7 m/s, while in segment 8 it is 7.2 m/s. At 3 s, the highest velocity is 9.2 m/s in segment 1 and decreases to the smallest velocity of 6.2 m/s in segment 8. It then increases to 8.4 m/s in segment 11. At 4 s, the velocity in segment 1 is 9.0 m/s and decreases to 8.3 m/s in segment 3. At 4 s, the smallest velocity is 5.3 m/s in segment 7 while in segment 8 it is 8.7 m/s. At 5 s, the velocity in segment 1 is 9.0 m/s and decreases to 8.3 m/s in segment 2. The smallest velocity is 5.1 m/s in segment 7. It is 8.8 m/s in segment 11 and decreases to 7.9 m/s in segment 12.

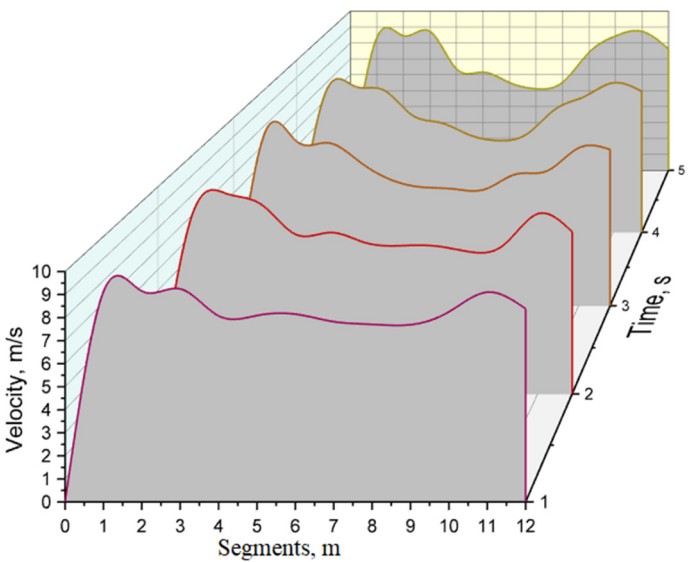

**Figure 9.** Average traffic velocity over distance and time.

Figures 7–9 show the temporal evolution of density, flow, and velocity, respectively. It is clear from these results that the proposed simulator can predict density, velocity, and flow at bottlenecks. Figure 7 shows the velocity for the corresponding changes in density in Figure 9. This is important in order to understand the critical density behavior and how it relates to velocity. Further, the flow can be used to estimate traffic capacity and predict infrastructure utilization and limitations.

### 4.5. Fundamental Diagrams

In this section, the fundamental diagrams with a distance headway of 2 m and 10 m are provided. These diagrams show the maximum flow and the variations in density and velocity. Figure 10a presents the initial flow versus normalized density when the data are first recorded in the simulator. The maximum flow is 2.5 veh/s, called the critical flow. Beyond this point the flow decreases and traffic becomes congested. Conversely, below 2.5 veh/s the flow is uncongested and the traffic is typically free flow. Over time, the road capacity changes as the critical density changes. With a 2 m distance headway, at 1 s the maximum flow is 3.4 veh/s at a density of 0.65 as shown in Figure 10b. At 2 s, the maximum flow is 3.5 veh/s at a density of 0.50, and at 3 s the maximum flow is 3.6 veh/s at a density of 0.60. At 4 s, the maximum flow is 3.4 veh/s at a density of 0.40–0.50, and at 5 s the maximum flow is 3.0 veh/s at a density of 0.32. Note that the maximum flow determines the road capacity.

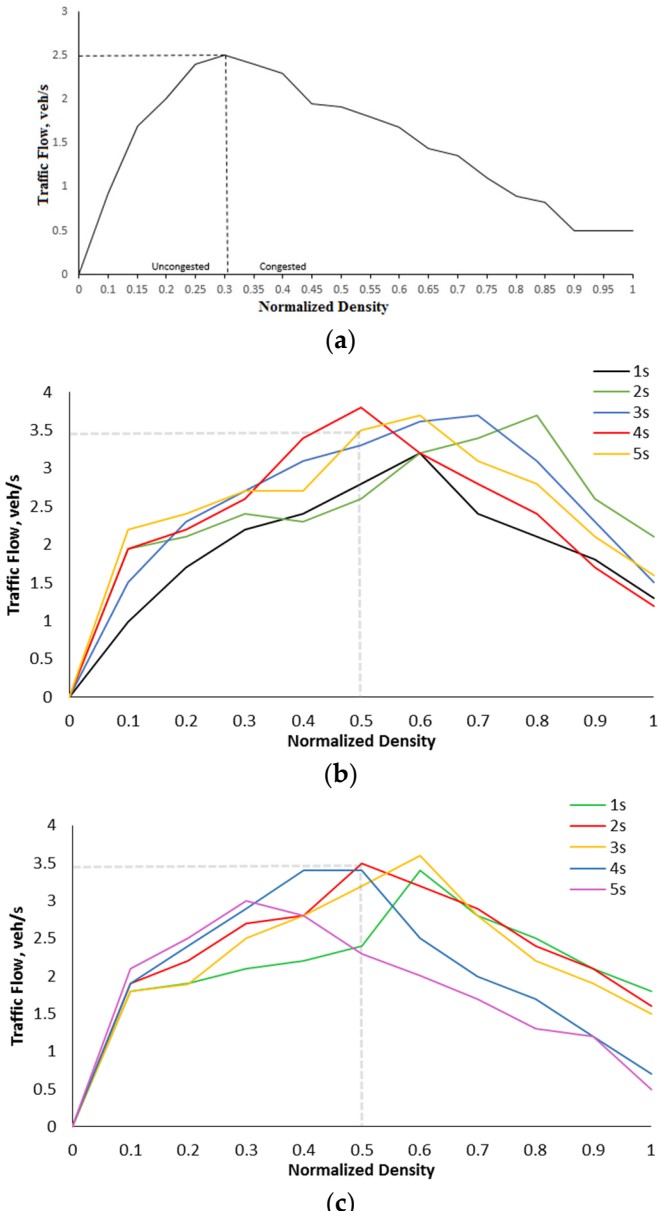

**Figure 10.** Fundamental diagrams between flow and density at (**a**) *t* = 0 s, (**b**) 2 m distance headway, and (**c**) 10 m distance headway.

With a 10 m distance headway, at 1 s the maximum flow is 3.0 veh/s at a density of 0.60 as shown in Figure 10c. At 2 s, the maximum flow is 3.6 veh/s at a density of 0.80, and at 3 s the maximum flow is 3.6 veh/s at a density of 0.70. At 4 s, the maximum flow is 3.5 veh/s at a density of 0.50, and at 5 s the maximum flow is 3.7 veh/s at a density of 0.60. Figure 10b,c show that the road capacity differs according to the density. These results provide insight for traffic planners to determine the road capacity based on the density with a given distance headway.

Figure 11a presents the initial velocity versus normalized density when the data are first recorded in the simulator. With a 2 m distance headway, at 1 s the maximum velocity is 9.0 m/s at a density of 0–0.30 as shown in Figure 11b. At 2 s, the maximum velocity is 10 m/s at a density of approximately 0 while the minimum is 6.5 m/s at a density of 0.70. At 3 s, the maximum velocity is 9.6 m/s at a density of approximately 0 while the minimum is 5.5 m/s at a density of 0.80. At 4 s, the maximum velocity is 8.6 m/s at a density of approximately 0 while the minimum is 5.0 m/s at a density of 0.80. At 5 s, the

maximum velocity is 9.0 m/s at a density of approximately 0 while the minimum is 4.0 m/s at a density of 0.80.

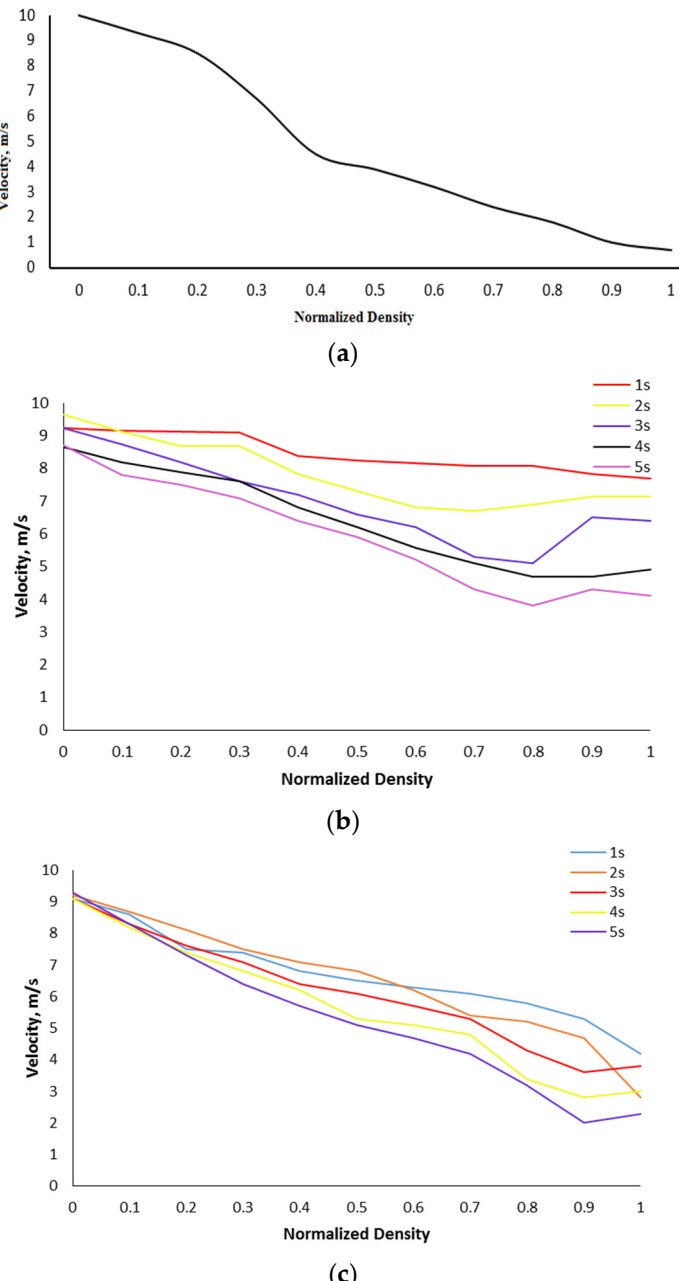

**Figure 11.** Fundamental diagrams between normalized velocity and density at (**a**) *t* = 0 s, (**b**) 2 m headway, and (**c**) 10 m distance headway.

With a 10 m distance headway, the maximum velocity is approximately 9.0 m/s at 1 s, 2 s, 3 s, 4 s, and 5 s, and occurs at a density of approximately 0, as shown in Figure 11c. At 1 s, the minimum velocity is 6.0 m/s at a density of 0.90, and at 2 s it is 5.0 m/s at a density of 0.90. At 3 s, the minimum velocity is 4.0 m/s at a density of 0.90, and at 4 s it is 3 m/s at a density of 0.90. At 5 s, the minimum velocity is 2.5 m/s at a density of 0.90. Figure 11b,c indicate that the velocity with a 10 m distance headway decreases more than with a 2 m distance headway as the density increases. These results can be used by traffic planners to estimate velocity limits on roads to efficiently utilize road infrastructure.

Figure 12a presents the initial velocity versus traffic flow when the data are first recorded in the simulator. The free flow velocity (uncongested traffic velocity) varies between 5.8 m/s and 10 m/s while the congested traffic velocity varies between 0 m/s and 5.8 m/s. The maximum flow is 2.5 veh/s. Figure 12b shows that with a 2 m distance headway, at 1 s the velocity is 8.5 m/s at a maximum flow of 3.3 veh/s. The uncongested velocity varies between 8.5 m/s and 9.5 m/s while the congested velocity varies between 7.5 m/s and 8.5 m/s. At 2 s, the velocity is 7.0 m/s at a maximum flow of 3.7 veh/s. The uncongested velocity varies between 7.0 m/s and 10 m/s while the congested velocity varies between 6.5 m/s and 7.0 m/s. At 3 s, the velocity is 5.5 m/s at a maximum flow of 3.7 veh/s. The congested velocity varies between 3.5 m/s and 5.5 m/s while the uncongested velocity varies between 5.5 m/s and 9.0 m/s. At 4 s, the velocity is 6.0 m/s at a maximum flow of 3.6 veh/s, and at 5 s the velocity is 6.0 m/s at a maximum flow of 3.7 veh/s. The congested velocity varies between 3.5 m/s and 5.2 m/s while the uncongested velocity varies between 5.2 m/s and 8.5 m/s.

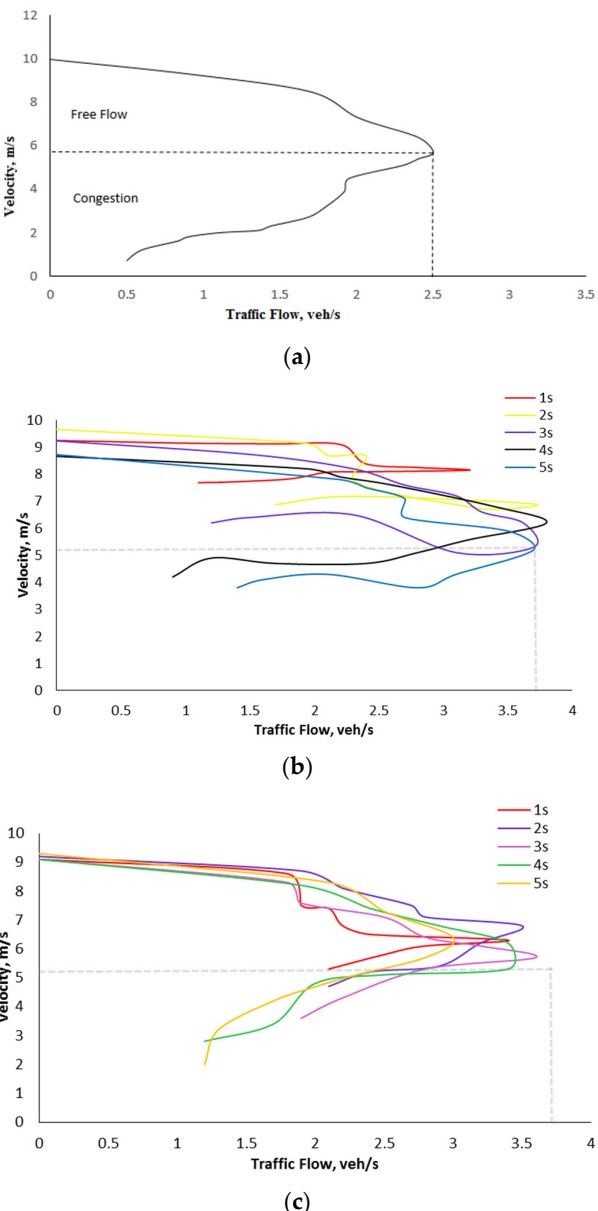

**Figure 12.** Fundamental diagrams between normalized velocity and flow at (**a**) *t* = 0 s, (**b**) 2 m headway, and (**c**) 10 m headway.

Figure 12c shows that with a 10 m distance headway, at 1 s the velocity is 6.5 m/s at a maximum flow of 3.4 veh/s. The uncongested velocity varies between 6.5 m/s and 9.0 m/s while the congested velocity varies between 5.2 m/s and 6.5 m/s. At 2 s, the velocity is 7.0 m/s at a maximum flow of 3.7 veh/s. The uncongested velocity varies between 7.0 m/s and 9.0 m/s while the congested velocity varies between 4.5 m/s and 7.0 m/s. At 3 s, the velocity is 5.5 m/s at a maximum flow of 3.5 veh/s. The congested velocity varies between 4.8 m/s and 5.5 m/s while the uncongested velocity varies between 5.5 m/s and 10 m/s. At 4 s, the velocity varies between 5.0 m/s and 6.5 m/s at a maximum flow of 3.5 veh/s. At 5 s, the velocity is 6.0 m/s at a maximum flow of 3.0 veh/s. The congested velocity varies between 1.7 m/s and 6.0 m/s while the uncongested velocity varies between 6.0 m/s and 9.5 m/s. The results in Figure 12 can be used to determine the maximum velocity and maximum flow. Further, with a 2 m distance headway the smallest velocity is 3.5 m/s at 5 s while with a 10 m distance headway the smallest velocity is 1.7 m/s at 5 s. This shows that the proposed simulator can help define velocity limits for road infrastructure.

The results presented in this section indicate that as the distance headway changes, the maximum flow changes. With a 10 m distance headway, there is a decrease in velocity as shown in Figure 11c, but with a 2 m distance headway the maximum velocity is maintained when the density at the bottleneck is small. Figures 10 and 12 illustrate how the maximum flow varies with 2 m and 10 m distance headways. This information can be used by personnel to control traffic at bottlenecks for efficient flow and reduced emissions.

## 5. Discussion

Traffic engineers and researchers can choose between several tools for homogenous traffic flow simulation. However, only two (SUMO and HetroSim) can simulate heterogeneous traffic flow [5]. Both SUMO and HetroSim have limitations. HetroSim can only consider small road segments and supports only a few vehicle types and speeds. SUMO is only suitable for lane traffic and enforces lane discipline on all vehicles including small two-wheel vehicles. This results in unrealistic behavior and poor road space utilization with heterogeneous traffic.

The macroscopic heterogeneous traffic flow simulator HetroTraffSim was developed to overcome the limitations of SUMO and HetroSim. As opposed to HetroSim, HetroTraffSim can simulate user-defined vehicle characteristics (including vehicle numbers, sizes, and speeds). Further, as opposed to SUMO, it does not enforce lane discipline. It employs a user-defined distance headway to determine vehicle overtaking. HetroTraffSim also employs raycasting along with the Unity3D engine NavMesh system for overtaking and avoiding collisions. HetroTraffSim can simulate traffic flow in a 3D environment to provide a more realistic look, unlike HeteroSim and SUMO. The low polynomial 3D models make HetroTraffSim computationally efficient so that simulations can run at 60 frames per second. The traffic simulations considered here were performed for 5 s, but much longer times are feasible. In the future, simulations will be performed over larger time durations under different traffic conditions. Further, the simulator will be extended to incorporate main arterial roads with actual traffic lane conditions and pedestrian facilities. This will allow the impact of pedestrians on vehicle flow to be evaluated. Finally, other heterogeneous traffic models will be added to the simulator.

## 6. Conclusions

Existing tools for traffic simulation consider homogenous and/or microscopic traffic flow or have limited capabilities. In this paper, a new Traffic Simulation Software (TSS) tool for macroscopic heterogeneous traffic flow was presented called HetroTraffSim. It was created using Unity3D and employs the second-order Khan–Gulliver (KG) model. This tool was evaluated on a bottleneck section of University Road, Peshawar, Pakistan. The inputs included road width, road length, and initial traffic density, and the outputs were velocity, flow, and density in each road segment. Traffic engineers can use this data for infrastructure planning and traffic management. HetroTraffSim is based on parameters that

can easily be obtained and has low computational complexity. The use of Unity3D means sensor data can be entered into the raycasting module for traffic automation. HetroTraffSim can easily be integrated with other tools. The results obtained show that traffic velocity, density, and flow are impacted by distance headway. This can be used to improve road infrastructure efficiency.

In the future, HetroTraffSim can be extended to include more input parameters. Further, it can be evaluated for other road scenarios such as T-sections, U-turns, roundabouts, and circular roads.

**Author Contributions:** Conceptualization, Z.H.K., T.A.G. and K.S.K.; methodology, K.S.K. and Z.H.K.; software, A.Z., M.R.U., K.S.K. and Z.H.K.; validation, A.Z., M.R.U., K.S.K. and Z.H.K.; formal analysis, A.Z. and M.R.U.; investigation, A.Z., M.R.U., Z.H.K., K.S.K. and T.A.G.; resources, Z.H.K., K.S.K. and T.A.G.; writing—original draft preparation, A.Z. and M.R.U.; writing—review and editing, K.S.K., Z.H.K. and T.A.G.; visualization, A.Z. and M.R.U.; supervision, K.S.K.; project administration, K.S.K., T.A.G. and Z.H.K. All authors have read and agreed to the published version of the manuscript.

**Funding:** This work was funded by the Higher Education Commission (HEC), Pakistan through the National Centre for Big Data and Cloud Computing (NCBC) at the University of Engineering and Technology (UET) Peshawar, Pakistan.

**Data Availability Statement:** Not applicable.

**Conflicts of Interest:** The authors declare no conflict of interest.

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
