# Peer review of "HetroTraffSim: A Macroscopic Heterogeneous Traffic Flow Simulator for Road Bottlenecks"

_futuretransp, doi:10.3390/futuretransp3010022_

Round 1
Reviewer 1 Report
The topic is interesting. The simulation tool, its implementation, and the description of the results are complex and well-prepared. But, the scientific soundness is weak. A broader literature review of other studies of similar problems is missing.
For example, using the keyword „road bottleneck” we can identify 11 papers in the Sustainability journal and 48 papers in all journals from the MDPI group, and using the keyword „heterogeneous traffic” 57 and 358 accordingly. Some of them should be cited.
On the other sides, the existing list of review is very poor and contain only Author’s own positions in general.
The conducted study should be compared with the literature (after the enlargement of this list as was mentioned above).
I identified some smaller problems:
In sections 2 and 3, the first paragraphs should be classified as subsections 2.1 and 3.1 (with appropriate titles). Next subsections should be renumbered, as exemplary 2.1 to 2.2, 2.2 to 2.3, etc.
The section „literature review” should be separated from section 1. This new section should contain more elements (as mentioned above).
The section „discussion” should be added (before conclusions) with connections and commentary about the enhanced literature review.
The last section, better entitled as „conclusions”, should be developed and contain also more general elements.
Author Response
Dear Reviewer,
Please find enclosed the document detailing changes made to the manuscript in light of your comments.
Thank you

Reviewer 2 Report
The manuscript described research conducted on traffic flow simulation of a road bottleneck in a street in Pakistan.
The main problem of the manuscript is that authors do not differentiate clearly which tools have been developed by themselves and which ones have been leveraged from previous work. It is paramount to clarify this.
Line numbering would make it easier for reviewers to identify issues in the text. I exhort the authors to include line numbering.
Abstract: This text focuses too much on statement of the research problem and too little on the findings and contribution of the study.
The authors should explicitly mention that in Pakistan driving is on the left.
Resolution of figure 4 is very low.
Can the authors identify heterogeneous traffic flow in Figure 4?
Figure 5 is not very illustrative as the number of lanes on each area can hardly be seen.
What traffic parameters were measured and reproduced in the simulation?
What is the traffic composition? Does it affect the results?
When the simulation begins, where does software generate vehicles? Only at the initial segment or all along the section?
How many simulations were launched, and which data were used in each?
A 5 second simulation is too short to draft conclusions.
Figure 7 and table 3 show the same data. One of them should be deleted. Preferably the table. Same applies to fig.8 and table 4, and fig.9 table 5.
Figures 7, 8 and 9 are accompanied by cumbersome explanations of the values measured. However, except in the case of speed graph, no clear patterns are observable in the data.
The conclusions related to headway and traffic management seem to be of negligible utility.
Do the authors have any countermeasure concerning road cross section to avoid the bottleneck?
Author Response

(The authors gave the same response as above.)

Round 2
Reviewer 1 Report
All of my remarks were considered. I am satisfied.
I have identified some little problems yet.
The references list and their numbering should be corrected. The order in the list must correspond with the moment of citation in the text.
Please, check the numbering of subsections in section 3. Now, two sections 3.1 exist.
Change the title of the last subsection to “Conclusions”. They are more than one conclusion there.
Author Response
Please find attached our response for your comments.

Reviewer 2 Report
The authors have addressed my comments to some extent. However, I consider that additional changes should be made before the manuscript is published.
As a response to one of my previous comments, the authors state that traffic parameters were measured and reproduced in the simulation, and that
heterogeneous traffic composition was considered, affecting the traffic forecast. This must be included in the manuscript.
The authors should also include in the manuscript that only one 5-second simulation was launched and state the limitations of this fact on software validation.
Finally, the authors should include in the manuscript their response about the future applications of their tool.
Please, for the next revision, indicate in the response to the reviewer’s comments the line where the changes in the manuscript are located.
Author Response
Please find attached our response to your comments.
Thank you

Round 3
Reviewer 2 Report
The manuscript is suitable for publication
Author Response
Dear reviewer, thank you for pointing out our oversights. The paper has been proof read by Prof. Thomas A. Gulliver who is a native English speaker. He is an author/co-authors of over 800 publications. All the mistakes and oversights have been corrected.
